# The impact of postoperative glucocorticoids on complications after head and neck cancer surgery with free flap reconstruction: A retrospective study

Tao Luo◉, Ren Zhou◉, Yu Sun◉*

Department of Anesthesiology, Shanghai Ninth People's Hospital, Shanghai Jiao Tong University School of Medicine, Shanghai, China

◉ These authors contributed equally to this work.
* dr_sunyu@163.com

## Abstract

### Background

After head and neck cancer surgery with free flap reconstruction, the use of glucocorticoids is often required to alleviate inflammation and edema. However, the impact of glucocorticoid on postoperative complications and cancer progression remains unclear.

### Methods

This retrospective cohort study included 711 elderly patients who underwent head and neck cancer surgery with free flap reconstruction at Shanghai Ninth People's Hospital from January 1, 2014, to December 31, 2022. Patients were categorized based on postoperative glucocorticoid usage into a high-dose steroid group (n=307) and a control group (n=404). The study focused on the impact of postoperative GC use on postoperative complications and long-term oncological outcomes.

### Results

Multivariate analysis indicated that compared to the control group, the high-dose steroid group had a significant increase in postoperative complications, including atelectasis (OR: 3.83, 95% CI: 1.27–14.11, P=0.025), postoperative hyperglycemia (OR: 1.54, 95% CI: 1.14–2.08, P=0.006), and flap complications (OR: 4.61, 95% CI: 3.31–6.47, P<0.001). These complications often required extended hospital stays (β: 1.656, 95% CI: 1.075-2.236, P < 0.001). Additionally, the high-dose steroid group had a higher rate of unplanned readmissions within one year (OR: 5.61, 95% CI: 3.87–8.25, P< 0.001). The increased readmission rates were notably due to difficulties swallowing requiring percutaneous gastrostomy (OR: 3.62, 95% CI: 1.97–6.98, P < 0.001), recurrence (OR: 9.34, 95% CI: 5.02–19.05, P < 0.001), and metastasis (OR: 4.78, 95% CI: 2.58-9.44, P < 0.001).

**Data availability statement:** Data cannot be shared publicly because of the hospital regulations. Data are available from the Ethics Committee of the Ninth People's Hospital Affiliated to Shanghai Jiao Tong University School of Medicine (contact via email: shjyiec@163.com), for researchers who meet the criteria for access to confidential data.

**Funding:** The author(s) received no specific funding for this work.

**Competing interests:** The authors have declared that no competing interests exist.

## Conclusion

The use of high-dose postoperative glucocorticoids is associated with increased post-operative complications, higher readmission rates, and poorer oncological outcomes in patients. The results advocate for cautious use and dosage management of perioperative glucocorticoids in head and neck surgeries to optimize patient outcomes.

## Introduction

Head and neck cancer (HNC), predominantly comprising squamous cell carcinoma (SCC), remains a significant clinical challenge due to its high incidence, mortality, and morbidity after treatment [1,2]. The complexity of treatment increases with the necessity for reconstructive surgeries such as free flap reconstruction (FFR) to restore functional and aesthetic deficits following extensive tumor resections [3,4]. Management strategies have steadily integrated advances in surgical, chemotherapeutic, and radiotherapeutic realms, nevertheless, improvements in prognosis and quality of life post-treatment are still greatly needed [5].

As surgical techniques advance, there is a growing recognition that the metabolic and systemic responses to cancer can influence treatment outcomes. In this context, glucocorticoids (GCs), while traditionally deployed for their potent anti-inflammatory and immunosuppressive properties, represent a dual-edged sword in cancer care [6]. GCs bear a particular significance due to their intricate roles in not only mediating stress response and metabolic modulation but also potentially affecting tumorigenic processes directly and indirectly [7–10].

Currently, the interplay between GCs, inflammation, cancer biology, and surgical outcomes remains inadequately understood. The variability in individual glucocorticoid synthesis and metabolic processing within the tumor microenvironment, modulated by enzymes such as 11β-hydroxysteroid dehydrogenase types 1 and 2, further complicates this challenge [11,12]. These differences could influence local and systemic effects, including immune modulation, glucose metabolism, and the inflammatory cascades pivotal for both cancer progression and postoperative recovery.

Recent studies underscore the pervasive activation of the glucocorticoid system in both regional tissues and systemic levels in patients with HNC. Such systemic elevation has been correlated with poor therapeutic outcomes and heightened corticosteroid-induced regulatory mechanisms that might foster tumor progression and resistance to therapies [12–14]. Moreover, emerging evidence suggests that the perioperative administration of systemic GCs could impact immediate surgical outcomes and increase higher short-term mortality in head and neck cancer patients undergoing FFR, potentially implicated by perturbed wound healing, and postoperative hyperglycemia [15–17]. However, these studies are limited by small sample sizes, and research on the impact of GCs on long-term complications remains scarce in patients undergoing head and neck cancer surgery with free flap reconstruction (HNS-FFR).

This study aims to refine our understanding of how exogenous glucocorticoid administration intersects with these complex biological processes, particularly in the setting of HNS-FFR. We plan to investigate the multidimensional impacts of GCs administered during the postoperative period in a retrospective cohort of head and neck cancer patients, focusing on their potential to alter postoperative complications, the rate of unplanned readmissions within a year, and the risk of cancer recurrence and metastasis which may be the first time in the in clinical practice. Such insights will be fundamental for optimizing therapeutic strategies and tailoring perioperative care to improve both oncologic and functional outcomes in this challenging patient population.

## Materials and methods

### Study design and participants

This retrospective cohort study enrolled elderly patients who were admitted to Shanghai Ninth People's Hospital, affiliated with Shanghai Jiao Tong University School of Medicine, from January 1, 2014 to December 31, 2022. The study adhered to the STROBE guidelines (see S1 Checklist ). Approval was obtained from the center's ethics committee (Ethics Reference: SH9H-2024-T315-1), and, due to its retrospective nature, the requirement for informed consent was waived. Data collection for research purposes began on September 2, 2024, following approval from the consent of the Ethics Committee. Authors had access to information that could identify individual participants during data collection. But prior to statistical analysis, information that could identify individuals was removed.

Inclusion criteria were: (1) aged 60 years or older, (2) diagnosed with a head and neck malignant tumor, and (3) underwent initially elective radical surgery such as maxillectomy, partial glossectomy, or partial buccal resection followed by free-flap reconstruction under general anesthesia.

Exclusion criteria included: (1) long-term use of GCs, (2) long-term use of immunosuppressants or being in an immunosuppressed state, (3) previous head and neck radiotherapy, and (4) missing data. Detailed registration information is shown in S1 Fig.

### Data sources and variables

Data were sourced from the Union Digital Medical Record Browser (Version 2012.4; Unionnet Co., Ltd., Shanghai, China) digital medical records. Extracted data underwent a dual-check process and data cleaning to eliminate outliers and duplicate records.

Collected demographic medical record information included gender, age, weight, height, smoking and drinking habits, American Society of Anesthesiologists (ASA) physical status classification, history of chemotherapy, hypertension, diabetes, chronic heart failure, and chronic obstructive pulmonary disease (COPD).

Surgical information collected encompassed the primary lesion location, pathological type, TNM stage, free flap site, lymph node dissection extent (none/unilateral/bilateral), intraoperative transfusion, and respiratory management after surgery (tracheotomy/tracheal intubation).

### Outcome and complication data

The primary outcomes of interest in this study were postoperative complications during hospital stay and unplanned readmissions within one year.

The postoperative complications collected included glycemic control (random blood glucose values ≥ 180 mg/dl) [18]; the Clavien-Dindo classification of surgical complications (S1 Table) [19]; respiratory system complications, including pneumonia, atelectasis, pulmonary embolism, and respiratory failure [20]; flap complications, such as flap infection, postoperative bleeding, flap dehiscence, seroma/fistula formation, partial flap necrosis, and flap crisis; and digestive system complications including liver dysfunction, acute pancreatitis, and intestinal obstruction. Other complications noted were cardiac insufficiency, cerebral embolism, acute kidney injury (AKI) (an absolute increase in plasma creatinine of 26.5 μmol/L within 48 h or a 1.5-fold increase in creatinine within 7 days) [21], deep vein thrombosis, postoperative delirium, and electrolyte imbalance.

Data on postoperative outcomes also included reasons for unplanned readmissions within one year, such as difficulty swallowing requiring percutaneous gastrostomy, lung infection, delayed healing at the surgical site, recurrence, and metastasis.

Secondary outcomes included length of ICU stay, postoperative hospital stay, unplanned reoperations, and mortality.

## Statistical analysis

Statistical analysis was performed using SPSS (version 26; IBM Corp., Armonk, NY, USA). A total of 560 mg of methylprednisolone was used as the cut-off value [22]. First, we measured and compared the incidence of each complication between the two groups. We then performed multiple regression to assess differences between different cohorts. Different regression analyses were used based on different types, using OR and β presentations respectively. Postoperative complications during hospital stay, unplanned reoperation, postoperative hospital stay, and unplanned readmissions within a year were compared between the two groups. The control group (less than the cut-off value) was used as a reference level for comparison. Age, gender, BMI, ASA score, history of smoking and drinking, and chemotherapy were taken as covariates. Subgroup analyses were performed according to cancer sublocus, pathological type, and T stage, and covariates were age, gender, alcohol consumption, diabetes, chemotherapy, end-organ damage (Clavien-Dindo classification grade 4), and blood transfusion. A two-tailed P value < 0.05 indicated statistical significance.

# Results

## Demographics

The study cohort comprised 711 patients, with 404 in the control group and 307 in the high-dose steroid group. Table 1 summarizes the demographic information of both groups. The groups were comparable in all respects except for the history of diabetes. The prevalence of diabetes was significantly lower in the high-dose steroid group compared to the control group (8.8% vs 24.8%, P < 0.0001).

## Surgical characteristics

The surgical characteristics of both groups are summarized in Table 2. Tongue cancer was the most common type of HNC (33.1%), and SCC accounted for approximately 90.6% of cases. The most common type of free flap used was the anterolateral thigh flap (50.0%). The high-dose steroid group was more likely to use this type of flap (55% vs 49.8%) and had a higher volume of intraoperative blood transfusion (P = 0.0347). Approximately 70.0% of patients underwent a prophylactic tracheotomy immediately after surgery.

## Outcome variables

The most common postoperative complications during hospitalization for patients undergoing HNS-FFR were pulmonary complications (43.3%), followed by digestive complications (37.7%), flap complications (34.5%), and electrolyte imbalances (34.5%) (see S2 Table).

Multivariate analysis (Table 3) revealed that, compared to the control group, the high-dose steroid group had a higher risk of postoperative complications (OR: 3.76, 95% CI: 2.11-7.13, P < 0.001), more difficulty in controlling blood glucose (OR: 1.54, 95% CI: 1.14-2.08, P = 0.006), and a higher incidence of atelectasis (OR: 3.83, 95% CI: 1.27-14.11, P = 0.025). Notably, the rates of flap infection and seroma/fistula formation were significantly higher (OR: 5.52, 95% CI: 3.79-8.16, P < 0.001; and OR: 4.98, 95% CI: 3.2-7.92, P < 0.001, respectively). Partial flap necrosis (OR: 1.96, 95% CI: 1.16-3.37, P = 0.013) and flap dehiscence (OR: 2.52, 95% CI: 1.17-5.75, P = 0.021) were also more common in the high-dose steroid group. These complications often necessitated longer hospital stays (β: 1.656, 95%CI (1.075-2.236), P < 0.001).

**Table 1. Baseline clinical characteristics.**

| characteristic | Control group (N = 404) | High-dose steroid group (N = 307) | P value |
|---|---|---|---|
| **Duration of use (day)**[b] | 4.27 (2.67) | 8.96 (3.18) | **<0.0001** |
| **Dosage of glucocorticoids**[b] | 294.26 (167.52) | 1213.94 (481.50) | **<0.0001** |
| Gender[a] | | | 0.6808 |
| Female | 148 (36.6) | 107 (34.9) | |
| Male | 256 (63.4) | 200 (65.1) | |
| Age[b] | 68.23 (6.44) | 68.74 (6.39) | 0.292 |
| Weight[b] | 62.19 (11.47) | 62.12 (10.58) | 0.9357 |
| Height[b] | 164.91 (8.00) | 165.63 (7.57) | 0.3485 |
| BMI[b] | 22.87 (3.45) | 22.80 (3.34) | 0.8556 |
| Smoke[a] | | | 0.666 |
| No | 273 (67.6) | 213 (69.4) | |
| Yes | 131 (32.4) | 94 (30.6) | |
| Drink[a] | | | 0.7935 |
| No | 306 (75.7) | 236 (76.9) | |
| Yes | 98 (24.3) | 71 (23.1) | |
| ASA[a] | | | 0.7874 |
| 1 | 23 (5.7) | 15 (4.9) | |
| 2 | 347 (85.9) | 265 (86.3) | |
| 3 | 33 (8.2) | 27 (8.8) | |
| 4 | 1 (0.2) | 0 (0.0) | |
| Chemotherapy[a] | | | 0.6801 |
| No | 335 (82.9) | 259 (84.4) | |
| Yes | 69 (17.1) | 48 (15.6) | |
| **Diabetes**[a] | | | **<0.0001** |
| No | 304 (75.2) | 280 (91.2) | |
| Yes | 100 (24.8) | 27 (8.8) | |
| CHF[a] | | | 0.6034 |
| No | 385 (95.3) | 289 (94.1) | |
| Yes | 19 (4.7) | 18 (5.9) | |
| COPD[a] | | | 0.7184 |
| No | 382 (94.6) | 293 (95.4) | |
| Yes | 22 (5.4) | 14 (4.6) | |

[a]: Data are presented as n (%) for categorial values;

[b]: Data are presented as mean (standard deviation) for continuous variables; BMI: Body Mass Index; ASA: American Society of Anesthesiologists; CHF: Chronic heart failure; COPD: Chronic obstructive pulmonary disease.

The high-dose steroid group also had a higher rate of unplanned readmissions within one year (OR: 5.61, 95% CI: 3.87-8.25, P < 0.001) compared to the control group. The increased readmission rates were particularly due to difficulties swallowing that required percutaneous gastrostomy (OR: 3.62, 95% CI: 1.97-6.98), recurrence (OR: 9.34, 95% CI: 5.02-19.05), and metastasis (OR: 4.78, 95% CI: 2.58-9.44) (P < 0.001).

Subgroup analysis indicated that the adverse effects of high-dose steroid use on non-planned readmissions within one year were consistent across different primary lesion location, pathological types, and T stages, with interaction P-values indicating no significant trends (0.309, 0.115, and 0.564 respectively, S3 Table).

**Table 2. Surgical information.**

| characteristic | Control group (N = 404) | High-dose steroid group (N = 307) | P value |
|---|---|---|---|
| Primary lesion location [a] | | | 0.5482 |
| cheek | 96 (23.8) | 77 (25.1) | |
| tongue | 134 (33.2) | 101 (32.9) | |
| gingiva | 83 (20.5) | 72 (23.5) | |
| others | 91 (22.5) | 57 (18.6) | |
| Pathological type [a] | | | 0.9697 |
| SCC | 365 (90.3) | 279 (90.9) | |
| ACC | 22 (5.4) | 16 (5.2) | |
| others | 17 (4.2) | 12 (3.9) | |
| T [a] | | | 0.1221 |
| 1 | 36 (8.9) | 26 (8.5) | |
| 2 | 200 (49.5) | 177 (57.7) | |
| 3 | 63 (15.6) | 45 (14.7) | |
| 4 | 105 (26.0) | 59 (19.2) | |
| N [a] | | | 0.0818 |
| 0 | 174 (43.1) | 160 (52.1) | |
| 1 | 120 (29.7) | 79 (25.7) | |
| 2 | 99 (24.5) | 64 (20.8) | |
| 3 | 11 (2.7) | 4 (1.3) | |
| M [a] | | | 0.5505 |
| 0 | 400 (99.0) | 306 (99.7) | |
| 1 | 4 (1.0) | 1 (0.3) | |
| **Free flap type** [a] | | | **0.0414** |
| anterolateral thigh flap | 201 (49.8) | 169 (55.0) | |
| radial forearm flap | 125 (30.9) | 69 (22.5) | |
| others | 78 (19.3) | 69 (22.5) | |
| CLND [a] | | | 0.3296 |
| none | 21 (5.2) | 9 (2.9) | |
| unilateral | 327 (80.9) | 254 (82.7) | |
| bilateral | 56 (13.9) | 44 (14.3) | |
| **Blood transfusion** [b] | 1.25 (1.55) | 1.51 (1.67) | **0.0347** |
| Respiratory condition [a] | | | 0.6825 |
| Tracheostomy | 288 (71.3) | 224 (73.0) | |
| Intubation | 116 (28.7) | 83 (27.0) | |

[a]: Data are presented as n (%) for categorial values;

[b]: Data are presented as mean (standard deviation) for continuous variables; SCC: Squamous cell carcinoma; ACC: Adenocarcinoma; CLND: Cervical lymph node dissection.

## Discussion

The understanding of the impact of corticosteroid administration on elderly patients undergoing HNS-FFR is limited. Therefore, in this study, we investigated the association between postoperative glucocorticoid administration and short-term postoperative complications, as well as unplanned readmissions within one year. The results indicate that while GCs may offer significant perioperative benefits, the use of high-dose glucocorticoids is associated with an increase in postoperative complications, poor metabolic control, and deteriorated oncological outcomes.

**Table 3. Multivariate Analysis of High-dose Steroid Group Compared to Control Group.**

| Complication | OR/β, 95% CI | P value |
|---|---|---|
| **Any complication** | 3.76 (2.11,7.13) | **< 0.001** |
| **Blood glucose control** | 1.54 (1.14,2.08) | **0.006** |
| Pneumonia | 1.3 (0.96, 1.76) | 0.091 |
| **Atelectasis** | 3.83 (1.27, 14.11) | **0.025** |
| Pulmonary embolism | 1.59 (0.34, 8.23) | 0.547 |
| Respiratory failure | 0.5 (0.13, 1.52) | 0.246 |
| Pulmonary complications | 1.3 (0.96, 1.76) | 0.091 |
| **Flap infection** | 5.52 (3.79, 8.16) | **< 0.001** |
| Postoperative bleeding | 1.87 (0.79, 4.62) | 0.156 |
| **Flap dehiscence** | 2.52 (1.17, 5.75) | **0.021** |
| **Seroma/Fistula formation** | 4.98 (3.2, 7.92) | **< 0.001** |
| **Partial flap necrosis** | 1.96 (1.16, 3.37) | **0.013** |
| Flap crisis | 0.94 (0.36, 2.36) | 0.892 |
| **Flap complications** | 4.61 (3.31, 6.47) | **< 0.001** |
| Hepatic insufficiency | 1.16 (0.85, 1.58) | 0.361 |
| Acute pancreatitis | 2.72 (0.26, 59.22) | 0.417 |
| Ileus | 0.74 (0.02, 20.58) | 0.839 |
| Digestive complications | 1.23 (0.91, 1.68) | 0.182 |
| Cardiac insufficiency | 0.88 (0.54, 1.43) | 0.619 |
| Cerebral embolism | 4.51 (1.07, 30.62) | 0.063 |
| AKI | 0.75 (0.15, 3.12) | 0.7 |
| Electrolyte imbalances | 0.99 (0.72, 1.36) | 0.931 |
| DVT | 1.07 (0.45, 2.53) | 0.871 |
| Postoperative delirium | 1.17 (0.74, 1.83) | 0.504 |
| Unplanned surgery | 1.49 (0.75, 3.02) | 0.256 |
| **PHS [a]** | 1.656 (1.075,2.236) [b] | **< 0.001** |
| **NPR** | 5.61 (3.87, 8.25) | **< 0.001** |
| **Difficulty swallowing** | 3.62 (1.97, 6.98) | **< 0.001** |
| **Recurrence** | 9.34 (5.02, 19.05) | **< 0.001** |
| **Metastasis** | 4.78 (2.58, 9.44) | **< 0.001** |
| Lung infections | 2.12 (0.91, 5.19) | 0.088 |
| Delayed healing | 1.94 (0.77, 5.14) | 0.165 |

[a]: Linear regression analysis;

[b]: β value (95%CI); AKI: Acute kidney injury; DVT: Deep vein thrombosis; PHS: Postoperative hospital stay; NPR: Non-planned readmissions within the first year.

Postoperative pulmonary complications (PPC) represent a significant issue in head and neck cancer surgeries, associated with high morbidity and mortality rates in patients undergoing HNS-FFR [23,24]. Our study observed that PPCs occur in up to 43.3% of postoperative head and neck cancer cases, aligning with previous findings [25]. Notably, patients in the high-dose steroid group exhibited a significantly higher incidence of postoperative atelectasis compared to the control group (OR: 3.83, 95% CI: 1.27-14.11, P = 0.025), and a trend towards statistical significance was also seen for pulmonary infections (P = 0.091). This increased susceptibility to infections such as pneumonia may be attributed to the immunosuppressive effects of high-dose glucocorticoids [26], which can exacerbate the immunodeficiency already present in patients with HNC [27]. Additionally, increased daily and cumulative doses of GCs

heighten the risk of skeletal muscle weakness, including the muscles essential for respiration and cough reflex [28,29]. This muscle weakness impairs the patient's ability to clear respiratory secretions, leading to mucus accumulation in the airways and resultant obstructive atelectasis.

Interestingly, we found that the high-dose steroid group was more susceptible to postoperative hyperglycemia (OR: 1.54, 95% CI: 1.14-2.08, P = 0.006), despite a higher prevalence of diabetes in the control group. This may be attributable to a higher proportion of patients with postoperative liver dysfunction in the high-dose steroid group (38.1% vs. 34.9%). Research shows that acute exposure to GCs can reduce the ability of adipocytes and hepatocytes to bind insulin, leading to insulin resistance [30]. Liver dysfunction undoubtedly exacerbates this phenomenon. Additionally, the cumulative dosage of steroids further contributes to the occurrence of hyperglycemia [31]. It is noteworthy that substantial observational evidence indicates that hyperglycemia in hospitalized patients, regardless of a diabetes diagnosis, is associated with adverse outcomes [18,32].

GCs are known for their anti-inflammatory properties, which may impede the inflammatory phase critical for wound healing. Successful progression and resolution of the inflammatory response are essential for proper wound healing. We observed that the high-dose steroid group had significantly increased risks of flap fistula formation (OR: 4.98, 95% CI: 3.2-7.92, P < 0.001), partial flap necrosis (OR: 1.96, 95% CI: 1.16-3.37, P = 0.013), and flap dehiscence (OR: 2.52, 95% CI: 1.17-5.75, P = 0.021). These findings highlight the detrimental effects of GCs on surgical recovery and wound integrity, consistent with previous studies [33]. This may be due to GCs interfering with various stages of the inflammatory response, preventing the smooth progression of each phase [34]. Moreover, GCs can also inhibit wound healing by suppressing the migration of keratinocytes [35]. Therefore, moderate use of GCs may be a crucial component in transitioning from inflammation to wound healing.

Patients receiving GCs appear particularly susceptible to infections [26]. Our study found that patients in the high-dose steroid group had a higher risk of flap infections compared to the control group (OR: 5.52, 95% CI: 3.79-8.16, P < 0.001). Kainulainen's study, which found an increased incidence of infections among oral cancer patients undergoing microvascular reconstruction treated with dexamethasone [15], seems to corroborate our findings. This could be due to high-dose corticosteroid-induced hyperglycemia increasing the risk of surgical site infections [17]. Additionally, delayed wound healing has been shown to elevate the risk of postoperative infections [33]. It is important to note that due to suppressed cytokine release, which reduces inflammation and febrile responses, patients on GCs usually have more challenges identifying infections early [36].

We also discovered that patients in the high-dose steroid group had significantly increased risks of recurrence and metastasis (OR: 9.34, 95% CI: 5.02-19.05, P < 0.001; and OR: 4.78, 95% CI: 2.58-9.44, P < 0.001, respectively), which is particularly concerning. The immunosuppressive properties of GCs act as a double-edged sword. While GCs can beneficially suppress the inflammatory responses that might exacerbate tissue damage during surgery, they can also inhibit the immune surveillance mechanisms necessary for detecting and destroying residual cancer cells [13,37]. This immunosuppression is especially significant in head and neck cancer patients, where the immune environment is already compromised by the tumor itself [27]. Additionally, glucocorticoid administration may exacerbate the immune evasion strategies employed by cancer cells, such as upregulation of immune checkpoints, known to facilitate immune escape in head and neck squamous cell carcinoma from host immunity [38]. Studies suggest that systemic GCs can enhance energy metabolism, potentially increasing the survival capacity of cancer cells during radiotherapy and chemotherapy both in vitro and in vivo [7,8,39]. This may be due to the fact that the sustained treatment with exogenous GCs activates

immunosuppressive transcription factors in dendritic cells, leading to the subversion of the anti-tumor immune responses induced by cancer therapies, and subsequent treatment failure [13,37]. Indeed, GCs have also been found to induce therapy resistance in solid tumors, including prostate, ovarian, and breast cancers [40]. Overall, the use of high-dose glucocorticoids postoperatively in head and neck cancer patients certainly warrants careful reconsideration.

## Limitations

This study has several limitations. Primarily, its retrospective design and single-center setting may limit the generalizability of the results. Fortunately, the database we used took effective measures to ensure data quality, and the sample size of our study was relatively large. Secondly, the inherent biases associated with retrospective analyses and the potential presence of unaccounted confounding variables may have affected outcomes. In addition, this study focused solely on unplanned readmissions within 1 year, excluding longer-term complications, which may have led to an underestimation of the true data and restricted our ability to assess the long-term effects and sustainability of the observed outcomes. Longitudinal studies with extended follow-up periods would provide more insight into the persistence of these effects over time. Future research should address these limitations by incorporating more diverse samples, improving data collection precision, and accounting for additional confounding variables. A larger sample size and multicenter designs would enhance the generalizability and reliability of the findings. Furthermore, prospective studies are needed to better define the causal relationships and to explore the mechanisms behind these observations.

## Conclusion

Postoperative overdose of GCs has a higher rate of complications, prolonged hospital stay, and an increased risk of recurrence and metastasis. In conclusion, while the perioperative administration of GCs in HNS-FFR can be beneficial for managing inflammation and stress responses, excessive use appears to compromise not only immediate surgical recovery but also long-term oncologic outcomes. These insights highlight the importance of tailored glucocorticoid regimens based on thorough preoperative assessments and intraoperative findings to optimize both immediate and long-term patient outcomes.

## Supporting information

**S1 Fig. Schematic flowchart.**
(DOCX)

**S1 Checklist. STROBE checklist.**
(DOCX)

**S1 Table. Clavien-Dindo classification.**
(DOCX)

**S2 Table. Postoperative data and complications.**
(DOCX)

**S3 Table. Subgroup analysis.**
(DOCX)

## Acknowledgments

Not applicable.

## Author contributions

**Data curation:** Tao Luo.

**Methodology:** Ren Zhou.

**Supervision:** Yu Sun.

**Validation:** Yu Sun.

**Writing – original draft:** Tao Luo.

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
