## [Decision Letter · Decision Letter 0]

13 Jan 2025

PONE-D-24-55543The Impact of Postoperative Glucocorticoids on Complications After Head and Neck Cancer Surgery with Free Flap Reconstruction: A Retrospective StudyPLOS ONE

Dear Dr. Sun,

Thank you for submitting your manuscript to PLOS ONE. After careful consideration, we feel that it has merit but does not fully meet PLOS ONE’s publication criteria as it currently stands. Therefore, we invite you to submit a revised version of the manuscript that addresses the points raised during the review process.

We look forward to receiving your revised manuscript.

Kind regards,

John Minh Le, MD, DDS

Academic Editor

PLOS ONE

Additional Editor Comments (if provided):

Reviewers' comments:

Reviewer's Responses to Questions

**Comments to the Author**

1. Is the manuscript technically sound, and do the data support the conclusions?

Reviewer #1: Yes

Reviewer #2: Yes

2. Has the statistical analysis been performed appropriately and rigorously? 

Reviewer #1: Yes

Reviewer #2: Yes

3. Have the authors made all data underlying the findings in their manuscript fully available?

Reviewer #1: Yes

Reviewer #2: Yes

4. Is the manuscript presented in an intelligible fashion and written in standard English?

Reviewer #1: Yes

Reviewer #2: Yes

5. Review Comments to the Author

Reviewer #1: 1. The topic is not unique but worthy of researching

2. There are many papers in google scholar and Refseek about this topic since 2020

3. Ethical approval and/or study registration is mentioned. Please clarify it in the method section

4. the title is attractive

5. the running title is not suitable

6. The abstract is clear. It is too long. You can shorten the background and the conclusion.

7. The aim is clear

8. The KEYWORDS are suitable

9. Lack of the graphical abstract

10. Lack of the abbreviations section

11. Lack of the highlight points

12. Arrangement of headings and subheadings are good

13. The introduction provides sufficient background information for readers in the immediate field to understand the problem/hypotheses

14. The paper arrangement is good

15. The method section is clear

16. The depth of the academic material is good

17. The study design is suitable

18. The suitability and accuracy of questions is good

19. The research methodology is reproducible

20. The materials are suitable

21. The inclusion criteria are good

22. The exclusion criteria are logical

23. The logic is clear

24. The presentation of the methods is good

25. Please submit a video abstract if it is available. It is an optional requirement

26. The paper is not novel. If it is novel, please clarifies how?

27. There are few grammatical errors in this article

28. The related concepts are introduced

29. The readability is sufficient

30. The results are interesting

31. The presentation of the results is good

32. The number of tables are adequate

33. Lack of figures?!

34. The structure of the tables are good

35. All tables are well referred to in the text

36. The theoretical analysis in this article is sufficient

37. The discussion of results from multiple angles is sufficient

38. The presentation of the discussion is good

39. The limitation of the study is clear

40. The conclusion is tenable

41. The recommendations are logical

42. The reference section contains many old ref

43. Please use (google scholar and Refseek) search engines then set it since 2020

44. The references are in order within the text

45. References are in one style

46. Bias is present

47. There is no conflict of interest with the author about this topic

48. Fund is mentioned

49. Conflict of interest is mentioned

50. Acknowledgement is mentioned

51. The significance of the paper is good

You can use my suggestions

My final decision is a minor revision

Reviewer #2: Hello. This is a good article and it explores an important topic. Please point out the limitations of the study, such as possible confounding factors and the lack of some information or the lack of accurate data recording in some cases. Also, make sure to mention any suggestions for future studies.

6. PLOS authors have the option to publish the peer review history of their article (what does this mean? ). If published, this will include your full peer review and any attached files.

**Do you want your identity to be public for this peer review?** For information about this choice, including consent withdrawal, please see our Privacy Policy .

Reviewer #1: **Yes: ** Hazim Alhiti

Reviewer #2: No

---

## [Author Response · Author response to Decision Letter 1]

18 Jan 2025

We thank the Reviewers and Editors for your constructive feedback to improve the manuscript and allow a resubmission to PLOS ONE. We have considered all the comments carefully and inserted our one-to-one responses. We believe that the updated manuscript has been substantially improved. Thank you in advance for your continued consideration. We have compiled our detailed responses to the editor's and reviewers' comments in a document titled "Response to Reviewers." For your convenience, we recommend reviewing the "Response to Reviewers" file for a better reading experience.

---

## [Editor Report · Decision Letter 1]

6 Feb 2025

The Impact of Postoperative Glucocorticoids on Complications After Head and Neck Cancer Surgery with Free Flap Reconstruction: A Retrospective Study

PONE-D-24-55543R1

Dear Dr. Sun,

We’re pleased to inform you that your manuscript has been judged scientifically suitable for publication and will be formally accepted for publication once it meets all outstanding technical requirements.

Kind regards,

John Minh Le, MD, DDS

Academic Editor

PLOS ONE

---

## [Editor Report · Acceptance letter]

PONE-D-24-55543R1

PLOS ONE

Dear Dr. Sun,

I'm pleased to inform you that your manuscript has been deemed suitable for publication in PLOS ONE. Congratulations! Your manuscript is now being handed over to our production team.

Kind regards,

on behalf of

Dr. John Minh Le

Academic Editor

PLOS ONE